# A Step toward Next-Generation Advancements in the Internet of Things Technologies

**DOI:** 10.3390/s22208072

**Published:** 2022-10-21

**Authors:** Farhan Amin, Rashid Abbasi, Abdul Mateen, Muhammad Ali Abid, Salabat Khan

**Affiliations:** 1Department of Information and Communication Engineering, Yeungnam University, Gyeongsan 38541, Korea; 2School of Electrical Engineering, Anhui Polytechnic University, Jiujiang District, Wuhu 241000, China; 3Department of Computer Science, Federal Urdu University of Arts, Science and Technology, Islamabad 45570, Pakistan; 4Faculty of Smart Engineering, the University of Agriculture, Dera Ismail Khan 29220, Pakistan; 5The College of Computer Science and Software Engineering, Shenzhen University, Shenzhen 518060, China

**Keywords:** data science, network science, Internet of Things, information science, social networks

## Abstract

The Internet of Things (IoT) devices generate a large amount of data over networks; therefore, the efficiency, complexity, interfaces, dynamics, robustness, and interaction need to be re-examined on a large scale. This phenomenon will lead to seamless network connectivity and the capability to provide support for the IoT. The traditional IoT is not enough to provide support. Therefore, we designed this study to provide a systematic analysis of next-generation advancements in the IoT. We propose a systematic catalog that covers the most recent advances in the traditional IoT. An overview of the IoT from the perspectives of big data, data science, and network science disciplines and also connecting technologies is given. We highlight the conceptual view of the IoT, key concepts, growth, and most recent trends. We discuss and highlight the importance and the integration of big data, data science, and network science along with key applications such as artificial intelligence, machine learning, blockchain, federated learning, etc. Finally, we discuss various challenges and issues of IoT such as architecture, integration, data provenance, and important applications such as cloud and edge computing, etc. This article will provide aid to the readers and other researchers in an understanding of the IoT’s next-generation developments and tell how they apply to the real world.

## 1. Introduction

The Internet of Things (IoT) is a promising research area that is based on the scenario of modern telecommunication [1]. The basic idea behind this concept is understanding the mechanism of communication of objects, things such as sensors, mobile phones, radio frequency identification (RFID), etc. These devices can interact and cooperate with neighbors to achieve their common goals. Obliviously, the development and the strength of IoT technology has a high impact on several aspects of people such as users and in our daily life. From the private user’s perspective, the effect of the IoT will be visible both in domestic and working fields. The domestic aspect assists in e-health, living, and enhanced learning. These examples show the possible application scenarios in which the new paradigm plays an important role. In addition, from the perspective of business users, the most apparent consequence is only available to the various research fields such as logistics, automation, business process, intelligent transportation of people and goods, etc. In the IoT, the most significant information is collected on the Internet. The current status and future growth of the IoT are shown in Figure 1 [2,3]. In this figure, IoT devices are compared to the world population and observe an increase year-wise. The IoT network generates a large amount of data collected via these devices. Therefore, these data double every two years and are expected to reach 163 zettabytes in 2025 [2]. It is observed that by 2025, the volume of data will increase ten times compared to the data generated in 2016. IoT devices require communication protocols, network infrastructure, and connecting technologies. The connecting technologies in the IoT are big data, machine learning, and artificial intelligence (AI), as shown in Figure 2. Big data are used for information gathering, analysis, and processing of large-scale data in the network [4]. The data are passed through a central controller, and the unique features of this technology are that it provides high efficiency, diversity, and also the capability to handle a large quantity of data. The diversity is empowered by using high efficiency, data types, and information processing [4]. The key advantage is to find useful information quickly and accurately from a large pool [4]. Artificial intelligence (AI) is a wide term and is based on computer programming and development. It is used to design and train machines and perform specific tasks [5]. AI is used to test theories such as consciousness and cognitive reasoning, etc. With the development of information technology (IT), the concept of data science becomes widely popular and has also become an important technology in the modern world. The objective is to find meaningful information and unseen patterns and to make intelligent business decisions. Generally, it uses machine learning models to build and predict. The network science discipline provides fundamental knowledge of complex networks [6]. The networks may be technological, but biological is also social. It might be possible that they are somehow a combination of all these types. With the development of IoT applications, a huge amount of data have been generated, and therefore the complexity increases. To overcome this issue, the integration of modern disciplines such as big data, data science, and network science and their connecting technologies is essential. As per our knowledge, there is very limited literature that has reported on this topic. For instance, a review of the integration of data science and network science tools and approaches was conducted by Amin et al. [7]. They highlighted the key aspects but did not focus on big data and AI. Wang et al. in [8] only covered the network science aspect of the IoT. Piccirilli et al. [9] presented an editorial note and explained the role of data science from an IoT perspective, but they did not explore the network sciences and big data. Similarly, Youhua et al. [4] presented the big data aspect and partially discussed the data science disciplines in their research study. Devi et al. [10] explored the data science discipline for the IoT. They explained the importance of data science from the perspective of the IoE (Internet of Everything) [11]. The IoE depends upon data science [10]. The authors did not cover network science and big data. Qiang et al. [12] presented a model for the implementation of a course system using data science and engineering but lacks the work on network science and big data. Similarly, Ranjan et al. [13] presented next-generation challenges regarding the integration of data science, the IoT, edge, and cloud. Foidl et al. [14] presented the challenges to improving the quality of assurance in IoT-based applications, but they did not review the most important aspects of big data and network science. Table 1 demonstrates the summary and the comparison of our literature survey and recent research studies. In this table, we examine that we have covered these disciplines and connecting technologies from all aspects. It is clear from this table is that very limited research has partially covered big data, data science, and the network science.

### 1.1. Objectives and Contribution

We present a comprehensive and systematic review concerning the recent advancements in the IoT from various disciplines and connecting technologies. The main goal of our review article is to provide a broad overview of the IoT; the integration from the perspective of big data, data science, and network science disciplines; and the relevant connecting technologies. Our critical research analysis is completely different from earlier studies since very few researchers have briefly/not covered the integration of these aspects, and we have covered it. We explored various connecting technologies such as big data, machine learning, and artificial intelligence and also discussed the strengths and weaknesses separately. The key contribution of our research is summarized below.

○In this article, we discuss an emerging paradigm named the IoT, including conceptual view, growth, distinctive features, core concepts, and basic functions.○We propose a systematic catalog that covers the most recent advances in the traditional IoT. We systematically cover state-of-the-art and recent studies in each aspect. We analyze and compare each study along with weaknesses and strengths and the application area. ○We discuss the network science discipline and the state-of-the-art studies that remained unexplored in the previous recent surveys.○We create a relationship between the IoT and big data management techniques at various levels such as collection, processing, analysis, and so forth. We discuss the importance of big data and its analysis concerning the IoT, complexity, key features, and the relationship with the IoT.○Our comparison and analysis differ from previous studies since most of the studies have partially or not covered these disciplines and relevant connecting technologies in detail. We performed a comprehensive analysis of machine learning, artificial intelligence, and blockchain as a solid solution to overcome the issues in the IoT. ○This article brings to light the current literature and illustrated their contribution to different aspects of the IoT. ○We highlight various research challenges that need to be identified, discussed, and addressed in the future. 

### 1.2. Paper Organization

The paper’s organization is as follows: Section 2 mainly provides the basics and the related work to this research area. The conceptual view of the IoT and its definitions are discussed in Section 3. The next-generation advances in the IoT are discussed in Section 4. Finally, we discuss the integration challenges such as integration, data province, data management, security, etc. 

## 2. Basics and the Background

In this section, we provide in-depth knowledge of the most recent research from the current literature and initiate a discussion on different approaches in the development of the IoT. The studies have shown a comparative analysis of the development of the IoT’s recent trends. For instance, Arzo et al. [15] conducted a survey on network automation for IoT technology. The basic philosophy of IoT technology is to connect everything in the world [15]. Therefore, it has changed human life. It has various applications such as smart homes, smart cities, smart transportation, etc. As the IoT end devices always communicate with others, measurement of values and the information that travels between the IoT devices is automated and without human intervention. The IoT connecting technologies are wireless fidelity (Wi-Fi) [16] and Lora [17], etc. As IoT technology helps us in automation, it is necessary to manage the network communication, interaction, and connectivity of IoT devices. The necessity of network automation arises especially when we consider the future network. The automation and efficiency are improved by using communication protocols, intelligent data-driven decision-making, etc. [15]. There are various authors worked on data-driven decision-making. 

For example, Sarker et al. [18] explained the concept of a smart city in their research study. At first, they explained the concept of a smart city. Then, they presented how a smart city system becomes more intelligent and also provides data-driven decision-making. They explored various machine learning and deep learning methods that are helpful to develop the analytical skills to fulfill the requirements of people in data-driven smart cities. The key insights of the paper are the integration of data science and the smart city. According to the authors, the key insights are thinking, conceptualization, and processing. Similarly, M.S et al. in [19] highlighted the concept of big data analytics while processing large sizes and the volume of data. Due to the growth of data, the volume and the complexity rise. To solve that issue, the authors highlighted the current state-of-the-art methods in big data and the use of advanced technologies such as artificial intelligence (AI). 

Marjani et al. [20] discussed the issue of big data that arises when an especially large number of IoT end devices communicate with each other. They discussed the state-of-the-art solutions in big data analytics and also the relationship between the IoT and big data. The authors proposed new architecture using an integration of the IoT and big data analytics. Ebert et al. [21] explored data science technology for the development of future software. At first, they explained the importance of data science as mandatory for today’s business. Data science facilitates the analysis, process, and management the valuable data [21]. This paradigm has become more popular, especially in convergence with the IoT domain [21]. Ebert et al. demonstrate how the software engineer benefits from the use of data science. Dalal et al. [22] presented a review of the recent machine learning algorithms for data science. They explained the integration of supervised learning [23], reinforcement learning [24], and unsupervised learning [25] as machine learning approaches and data preparation and data visualization as data science methods. Wu et al. [26] presented a useful approach for smart transportation and smart factory. They used basic data science concepts such as data collection and data analysis and suggested them for improving productivity and efficiency in business. They also highlighted various applications of the industrial Internet of Things (IIoT) [27] such as smart transportation, etc. Devi et al. in [10] highlighted the importance of data science as a sub-domain of the IoT. The authors highlighted traditional data science and the IoT. They proposed classification and a few recent solutions for the IoT, but they did not discuss the network science for advancements in the IoT. After careful analysis of the existing literature, we concluded that most researchers focused on the integration of big data, data science, or network science disciplines. None of them suggested using both disciplines as a future advancement for the IoT. In the next few sections, we first present the conceptual view of the IoT and distinctive features, and then we explain the basics of all disciplines and connecting technologies for the IoT. 

## 3. Conceptual View of the IoT

In the modern world, the IoT has become popular, and it has an impact on people’s daily lives [4], as well as the high potential to positively impact technological advancement in the IoT. The physical objects are connected to the Internet using sensors and actuators [1], as shown in Figure 3. There are numerous applications of the IoT, such as smart cities/transportation, smart homes/buildings, smart factories, smart healthcare, etc. All these IoT applications perform a very important role in improving the quality of our lifestyle. Figure 3 illustrates the conceptual view of the IoT. In this figure, the IoT end devices such as sensors and actuators are connected to physical objects. These devices send and receive data to IoT cloud servers through different communication technologies referred to as IoT access networks/gateways [28]. Generally, in this network, wireless communication technologies are the backbone of the IoT systems. These can enable connectivity between different machines and different applications. Examples of these communication technologies are Wi-Fi IEEE 802.11 [29], Bluetooth IEEE 802.15.1 [30], IEEE 802.15.4/802.15.4e [31], ZigBee [31], IEEE 802.11p [32], and WAVE [33]. The IoT end devices are extremely sensitive to the utilization of energy. Additionally, the execution of the applications of IoT systems must rely on the network system’s lifetime. These end devices forward data to the IoT cloud server using transit networks such as IPv4 [34], etc. The cloud server provides data storage, processing, and also analytics management. The user view provides various applications such as smart cities and smart factories [35]. The traditional methods for data processing need to be modified for efficiency purposes. Therefore, the development of new and highly efficient cross-domain technologies is necessary for the researcher’s community. In the next section, we discuss the integration of big data, data science, and network science disciplines and the integration advantages in detail. 

## 4. Next-Generation Advancements in the Internet of Things (IoT) Technologies

In this section, we describe big data, data science, and network science disciplines, and then we discuss the recent advancements in this area. We explore various connecting technologies such as machine learning, artificial intelligence, and federated learning and suggest next-generation advancements in the IoT. In addition, we discuss the most relevant recent literature along with the advantages and disadvantages. 

### 4.1. Big Data

In this section, we explain the concept of big data, its advantages, the five Vs, and the recent growth trends for the readers. The IoT end devices generate a large amount of data in the network. Gathering, analyzing, and managing a large amount of data requires advanced programming skills and statistical methods [36]. The data are generated from the IoT connecting end devices. The second step is to perform an analysis, and the third step is to manage the existing data. As different types of redundant and noisy data have been generated through a network, the IoT end devices degrade their performance with time. The key advantage of big data is to reduce the error of risk and also provide accurate data [36]. Big data provides five diverse dimensions sometimes known as the five Vs, as shown in Figure 4. In this figure, the volume refers to the size of big data. Generally, the data are known as big data or not—it mainly depends upon the volume. A recent survey [37] presents data growth dynamics and prediction trends, as shown in Figure 5. In this figure, we examine that the size of data has increased in the last 16 years. It will dramatically increase in the coming years and is forecasted to reach 175 zettabytes in 2024~2025 [37]. The global big data market is forecasted to grow to 175 billion U.S. dollars by 2025, more than double its expected market size in 2018. This rapid growth in volume and the size of data are due to the use of modern technologies such as cloud computing and edge computing, etc. [37]. The term “velocity” refers to the speed at which data are gathered in the network [12,38]. This is due to social media and mobile data, etc. The term “verity” refers to the type of data generated by either machines or humans such as unstructured, semi-structured, and structured data. The term “veracity” refers to the accuracy, credibility, integrity, and quality of data. Because the data are generated from different sources, we need to check these data before processing. The last feature of big data technology is “value”. The value refers to how the data are useful in the decision-making process. In this regard, we need to extract the value of big data using proper data analytics.

#### Recent Advances in Big Data

In this section, we explain the holistic view of the IoT and the most recent state-of-the-art research. Figure 6 explains the holistic view of the IoT and big data, as well as where the IoT end devices are connected [20]. IoT end devices such as CCTV cameras, smart home devices, and smart traffic light sensors are connected via the cloud. These devices carry a large amount of data named big data [39]. The data have different formats. Initially, the data are stored in a low-cost storage media. The storage is powered by cloud technology [40]. These data have been gathered using big data analytics, and the big data operations have been performed by using the five Vs. These data have been stored in large storage devices such as clouds, etc. The last step is to perform an analysis of data using tools such as Spark [41], Mapreduce [42] and Skytree [43], etc. These tools perform the analysis of data. The issue that occurred during data collection is the use of IoT technology along with intelligent software [36]. The intelligent data pre-processing techniques are meta-data creation and pre-processing [44]. The IoT end devices forward data to different locations and are then processed. As the data size is very large, it needs to be sent quickly to processing in different locations. If there is any latency, then there is a possibility of a valuable loss of data. Various researchers devoted work to this end; for instance, Assahli et al. [45] presented a comparative study on the cleaning and the pre-processing of IoT data. To do that, they suggested multi-agents for the pre-processing of IoT data. According to them, the agents help to reduce the complexity. They compared several studies related to the cleaning and processing of IoT data. From critical analysis, they found that multi-agent systems are very helpful for data pre-processing. The use of an agent helps to reduce the complexity. Similarly, Liang et al. in [46] suggested a deep learning-based model for fast data processing in IoT systems. The authors proposed a model. This model is very smart and highly efficient in computation. They identified that deep learning provides support for real-time applications. It helps in many ways. For instance, it helps to increase energy efficiency and also the spectrum in the IoT. In this study, the authors suggested applying the singular-value decomposition (SVD)-QR algorithm to the pre-processing of deep learning for large-scale data. 

Diyan et al. [47] proposed an energy management and processing model for the IoT. The necessity of electrical energy, renewable energy, and the smart grid introduces a new paradigm and extended the current form of electrical Energy Data Management and Processing Systems (EEDMS). To overcome this problem, they presented a multi-tasking and scalable solution named the Internet of Things Gateway (IoTGW). They implemented their proposed model and prepared a test bed using a real-time IoT data streaming network. Baker et al. in [48] discussed a special medical emergency when delays in communication could lead to possible detrimental effects on patients. In a few possible situations, the patient data for a particular event are required for processing. The pre-processing helps to reduce the huge volume of data in the network [36]. Moreover, it helps to move the sensor data using the processing function, and hence, the amount of data has been reduced. Modern IoT sensors have some built-in functions that help to perform pre-processing before the data submission. The intelligent data processing methods along with contributions for the IoT are listed in Table 2. Antonini et al. in [49] presented a unique model for audio sensors. These smart sensors work locally and can perform various computations using raw audio streams before transmitting using gateways. The second way is to create meta-data. After completing the first step, the next step is to store data and used them again. The meta-data are used to put the stored data into context. The stored data have been queried for information whenever it is necessary. It can require a lot of resources to process the data again due to the variety and volume. The meta-data help to speed up the process by adding some additional data, which describe the stored data. For instance Dobson et al. in [50] presented a reference model for the sensor and data warehouse. They employ advanced data warehousing methods to create an extendable model. This can be used to capture the contextual meta-data with sensor datasets and facilitate analysis of such datasets long after they have been collected [50]. Several researchers worked on the integration of big data and the IoT. For example, Mehdi et al. in [51] presented a survey using deep learning (DL) for the integration of the IoT and big data. In this survey study, they discussed the importance of DL to achieve the desired data analytics. They also discussed various DL models and algorithms for the integration of IoT data analytics. In addition, there are various algorithms and models on recent technologies such as fog and cloud centers in support of IoT applications that have been discussed in detail. Sollins et al. [52] presented a study on big data security and privacy in the IoT. Because the data are at the heart of the IoT [52], the issue of security and privacy in the context of the IoT arises, especially in the case of the collection and management of big data. In this research study, they highlighted the security and privacy aspects of data in three dimensions. The first is social policy context [53]. The second is the business and economics context, and the third is the design and technology context. In this study, they explored various authors’ works. Guo et al. [54] discussed the reliable traffic monitoring model based on the blockchain for vehicular networks [55]. In these networks, real-time traffic monitoring is a fundamental problem. This problem arises especially in the building of smart city applications [54]. To solve this problem, the authors proposed a reliable and efficient model. They design a lightweight blockchain framework to monitor the interactions between traffic administration and vehicles. The blockchain guarantees efficiency and reliability and security during execution. Their proposed utility function helps to automate the vehicles’ actions. Similarly, Guo et al. in another research study [56] proposed an architecture model for the multiple energies trading problem using byzantine “blockchain”. To handle the reduced latency and improve throughput, they introduced a novel credit model. The main contribution of this study was proposing this new credit model. The census is achieved by using a sum of the credits of voting nodes and expecting less than the number of voting nodes [56]. Kim et al. in [57] proposed an anomaly detection system based on blockchain networking. The blockchain provides strong cryptograph protection for the IoT-based systems. For this purpose, they propose a security model. This model can perform the analysis of blockchain network traffic and perform the statistics to detect malicious events through anomaly detection and data collection. The proposed anomaly detection engine detects the abnormal anoles from the created data instances based on the semi-supervised learning. 

### 4.2. Data Science 

In this section, we first explain the data science discipline and then explain the recent advances along with advantages. As we know, IoT technology has changed daily life and also modified the face of today’s business [58]. It has turned individuals into smart devices and connected businesses and consumers into overlapping enterprises. Data science is a multi-disciplinary subject and therefore draws from various fields of study such as data mining, machine learning, and distributed computing, as shown in Figure 7. In this figure, it is clear that this field of research has applications in various disciplines. Data science helps to process and supports the derivation of valuable data [59]. Figure 8 illustrates the data science lifecycle [12,60]. The lifecycle is based on the seven iterative steps such as data cleaning, data exploration, and data mining, etc. The data science lifecycle is helpful when using machine learning and analytical strategies to produce insights and also to predict information to acquire commercial enterprise objectives. 

#### 4.2.1. Recent Advances in Data Science 

In terms of the “IoT”, data refer to information that is generated by the end devices such as sensors, applications, some other gadgets, etc. [13]. Generally, data science uses data from both IoT systems and some other technologies. Then it transforms the data using analysis and visualization into value-based, decision-based systems. This process will help an organization or business meet the requirements and stay forward of their competitors in all aspects. In addition, the components of data science allow a deep understanding of values from the IoT and its development. Business heads and corporate leaders discovered the value of data collecting and analysis to increase their business performance and productivity. It is on these terms that data science grabbed the corporate world by storm. For instance, Qiang et al. in [12] discussed the importance of data science in engineering technology. By using key concepts of data science, the authors proposed a curriculum system for a university. This university curriculum system clarifies the relationships between the different courses. Ranjan in [13] presented the concept of the integration of data science and IoT technology. The authors discussed emerging technologies such as edge, cloud, and the IoT in detail. They explored the key challenges such as security and privacy, etc. Piccirilli et al. [59] presented a guest editorial on data science and the IoT. In this note, they explained the connection between data science and the IoT. They highlighted a few modern methods such as edge and machine learning, etc. Sajid et al. [61] highlighted the key applications of data science for the future industry 4.0 [62]. According to the authors’ recent data science methods and applications apply to every field, the experts and the researchers are always thinking about the root causes of failures and the quality deviations of machines. In this regard, the integration of data science and industry 4.0 will increase the efficiency and will be helpful predicting the quality of material by minimizing the time and production cost. They identified five critical processes of data science such as knowledge-based, data-driven, physical model, and digital twin approach for predictive maintenance. 

#### 4.2.2. The Connecting Technologies

In this section, we highlight the importance and the advantages of key connecting technologies, for instance, artificial intelligence and machine learning. Smart objects are connected via the Internet and, hence, form cyber–physical systems (CPSs) [63]. The concern that arises is how to handle a huge amount of generated data with lower computation power [64]. Researchers of data science and artificial intelligence are motivated to answer this question. The common functionality of the IoT and AI is shown in the Figure 9. This figure explains that emerging technology is helping to improve the accuracy rate and also operational efficiency. In general, AI is the ability of intelligence in machines to perform any tasks without human intervention [64,65]. The integration of AI and the IoT has tremendous prospects in today’s life [66]. Another recent technology is machine learning (ML). ML is a method to perform computation intelligently. Figure 10 explains the basic principle of ML functionally and integration with the IoT [67]. In this figure, we examine whether IoT sensor data are either structured or unstructured for the temperature or health sensors. Advanced ML methods have been applied to that data. The output useful applications are data analysis and business models. ML modules are helpful to find the hidden insights of sensed data in the IoT [68]. In addition, it is also used to predict the future requirements of individual users, governments, businesses, as shown in Figure 10. The goal of the IoT is to perceive what is happing in our surrounding environment and allow the automation of decision-making using intelligent methods [68]. The IoT aims to understand what people want and how people think, predict unwanted events, and learn and manage certain situations. The IoT needs to understand the data which have been processed by millions of objects. To do this, ML algorithms play an important role in the IoT [68]. The IoT technology is ubiquitous by nature, which means that it is available everywhere. ML plays an important role in digging out the data produced by the millions of connected devices [68]. The trends and tendencies of ML are used to find the patterns and may be the underpinning for human-like intelligence. The objective of ML in the IoT is to bring complete automation by enhancing the learning facilities’ intelligence through smart objects. For instance, fire prediction in an industrial or house kitchen by using an alarm sound is used to prevent fire. It is possible by using ML methods with the integration of IoT applications. For instance, a new and efficient framework has been proposed by combining ML and big data processing for the security of the mobile Internet of Things (MIoT) [69]. In this framework, the authors proposed five ML methods to solve the classification problem [69]. Hossain et al. [70] presented a comprehensive review paper on the integration of big data and ML for security in smart grids. The IoT smart grid provides efficient load forecasting and data acquisition methods with cost-effective solutions. Chaabouni et al. [71] discussed the most recent intrusion detection techniques in the IoT using ML. They analyzed the state-of-the-art proposal in the IoT in terms of deployment of algorithm, validation, and detection methodologies. In addition, they critically analyzed the ML and traditional techniques. Along with leading technologies, the blockchain technology is used to boost the ubiquitous IoT [72]. It has a variety of features such as information sharing, de-centralization, privacy protection, information traceability, etc. Therefore, it is suitable for IoT device constructions. Another emerging technology is federated learning (FL) [73]. It is a distributed ML technique that trains an algorithm across multiple de-centralized edge devices or servers holding local data samples without exchanging them [74]. This technique enables device training, keeps client local private data, and further updates the global model based on local module updates. To attain a better ML model under the conventional centralized approach, user security and privacy is compromised; in contrast, the FL approach is capable of training the model and leveraging the private data of clients without ever sharing them with other entities. Imteaj et al. in [74] presented a survey on FL especially for the resource-constrained IoT devices. In this research study, they explored various existing studies on FL and relative assumptions for distributed implementation using IoT devices. They highlighted the problems of state-of-the-art algorithms, especially when applying during FL within a heterogeneous environment. 

### 4.3. The Network Science 

In this section, we first explain network science and then we explain the recent advances in this area. In the IoT, end devices generate a large amount of data over networks; therefore, the efficiency, complexity, interfaces, dynamics, robustness, and interaction need to be re-examined on a large scale. Researchers should understand the user demands, network performance bound, the capacity of IoT networks, network infrastructure, etc. This will lead to seamless network connectivity and the capability to provide support for the emerging applications of IoT users. Briefly, the research on network science and engineering for IoT-based applications is still in its infancy [8]. In this section, we discuss the recent development and the future advances in engineering and network science that can lead to future emerging IoT applications. In many complex systems, there is a network that defines the interactions between the components. Network science is an emerging paradigm and a new discipline in the 21st century. It originated from the social network and graph theory [7]. In the words of Stephen Hawking, “I think the next century will be the century of complexity” [14]. In this research field, there are many theories, methods, and approaches derived from social science, computer science, mathematics, and graph theory. Therefore, this subject is known as a multi-disciplinary field [7]. 

#### The Recent Advances in Network Science 

The network enables us to address the complex issues regarding inter-dependent systems including those in the human, biological, economic, and other domains [75]. The IoT is based on the extension of the Internet. Thus it inherits Internet services and a complex structure [76]. The complex network theory is a mature field and also applies to different research fields such as social networks, the Internet, the World Wide Web, and cellular networks [76]. In this regard, scale-free networks are one good classic example [77]. This network is constructed based on the complex network topology [78]. The node’s degree follows the power law and is also based on preferential attachment (PA). There are few research studies which have discussed the similarity between the IoT and scale-free networks. For example, Wu et al. [79] discussed the IoT as a complex network. Complex networks approach the representation, characterization, and analysis of interactions between an enormous number of sensors, actuators, and processors in a systematic manner [79]. In this theoretical study, they discussed a few recent trends in analyzing and modeling the IoT systems with complex networks to provide an overview of IoT technologies that reside in networks and sensors to provide applications and services. They made an effort to develop an innovative, inter-disciplinary framework for the IoT. Batool et al. in [80] presented agent-based modeling [81] using complex networks [82] by utilizing the methods previously under the cognitive agents-based computing framework. The authors proposed a distributed algorithm using self-organization for the dynamic approximation of power utilization. Atov et al. in [83] presented the research on the current challenges related to IoT applications and the composition of data processing in infrastructure. They presented a comprehensive discussion on artificial intelligence (AI) and also suggested a new operating system to integrate the system libraries and the IoT systems. Li et al. [84] proposed a dynamic malware propagation model for the IoT. In this model, the devices are classified into different groups according to the level of dissemination. Zhang et al. [85] presented a complex network-based growth and evolution model (GEM) for the IoT. The GEM model has some special properties in terms of evaluation, growth, and preferential attachment. The problem is the fixed number of nodes at a certain time. 

## 5. Future Challenges in the Emerging Technologies 

The future advancements and challenges are given below. 

### 5.1. Integration

Future research efforts need to be carried out for the development of innovative methods and techniques to provide powerful support for the integration of heterogeneous data. However, the relationships among major IoT data sources such as CCTV [86], events (flooding), air pollution, and the stakeholders (first responders) are generally very difficult to detect. It mainly depends upon the specific context of the IoT application [13]. For instance, road traffic patterns, associated noise, and air pollution data must be connected via data integration approaches to provide better air quality monitoring. The diverse types of data have been extracted from water quality sensors, traffic flows, and sensors, for example, CCTC, etc. The diverse data integration methods are very helpful to improve the current standards such as semantic sensor networks (SSN) [87]. These standards provide support for the consistent representation of IoT sensors. A new and interesting research direction suggests the use of graph and machine learning methods for the discovery of relationships between associated events, IoT data sources, and also stakeholders [13], etc. 

### 5.2. Data Provenance

The data provenance [88] method is used for logging the origins of data, for example, IoT data sources [89]. The different data provenance methods used for IoT sensing are helpful to improve the overall decision-making process [13]. Generally, the decision systems such as traffic control systems require meta-data with the aim to validate the post-event decisions. This will also provide aid for improving the overall efficiency and reliability in the process of decision-making. Therefore, real-time decision-making systems become more and more complex. In this way, the IoT meta-data and the data provenance become more critical in providing the implementation. The other elements such as social use and trust are very essential components for the development of IoT sensing systems [90]. The challenging task is to deal with provenance in terms of handling large IoT graphs. It is due to the data, volume, size, and also configuration such as edge or cloud. Therefore, the researchers should focus on the development of contextual meta-data. The traditional data provenance methods acquire large data volumes and therefore are not applicable for modern IoT applications. New methods are required that are helpful to reduce and enhance the data provenance, transmission, and meta-data collection. One possible future research direction is to use blockchain methods such as blockchain-distributed technology (DLT) [13]. This technology is used to record the data lifecycle activities [91]. The second key challenge is handled by introducing the new data provenance techniques, and, hence, data privacy is insured.

### 5.3. IoT for Data Management, Security, Privacy, and Compliance 

Data security is the main concern that arises, especially as the connected end devices are increasing in the IoT network [15,92]. The stakeholders and users are key components of IoT systems [93]. They implicitly expect the security and privacy of their valuable data [13]. The IoT end devices are exposed to limit the user’s privacy. Organizations need to take care of preventing and avoiding the leakage of valuable data. In recent years, data science methods play an important role in providing the security of users. The security in IoT applications involves two main properties such as integrity and authentication. To provide the security feature in the IoT, two important conditions must be met: one is suitable authentication, and the second is the device identity. The IoT devices are diverse and have different vendors, and they also have no central entity. This makes up the traditional enterprise, which is based on the management of systems. The key concern for the modern world is to further focus on the development of modern methods to identify the IoT devices working, especially in distributed or de-centralized systems [94]. As the traditional IoT models are no longer applicable to the IoT systems due to constraints such as battery life, memory, etc., researchers should identify the advances in IoT frameworks. Another aspect is to improve security by using recent technologies such as ML, AI, blockchain, etc. [67]. To handle a large amount of data for processing, ML techniques can be combined with the IoT. It would be helpful to improve the accuracy rate and the operational efficiency rate. The ML techniques consist of supervised and unsupervised learning. The ML addresses security issues such as intrusion detection, anomaly detection, malware detection [95], false data injection [96] and impersonation, unauthorized IoT device identification, etc. [67]. The authors in [64] highlighted the integration of AI in the IoT. The AI-enabled IoT to improve the accuracy rate helps in predictive analysis and increases overall operational efficiency [67]. The AI also provides some security regarding issues such as authorization, privacy preservation, malware detection, etc. Blockchain technology is the most promising technology that helps address identity verification, trust management, secure storage and computation, data integrity and secure communication, access control, information sharing, firmware detection, self-healing, etc. [67]. Another leading technology is known as FL. It is very helpful for resource-constant IoT devices. It helps to improve scalability and data security [74].

### 5.4. Unified Messaging and Scalability

Generally, IoT systems use hybrid methods for message passing [13]. It may depend upon the specific context whether it is de-centralized or centralized. The message communication includes a sensor to the cloud (S2C) [97], a sensor to the edge (S2E) [98], and an edge to the cloud (E2C) [99]. These centralized modes, for example, S2C, cannot provide support for real-time applications. In addition, the current protocol such as Azure IoT [100] is a cloud-based protocol and is unable to meet the quality of service requirements for the IoT systems. Therefore, the development of new communication protocols needs to include the edge and cloud layers. The data scale is the major concern of the IoT. Generally, the IoT end devices generate big data every second, so organizations should implement a better approach to logging big data [101]. The new models and communication protocols for the IoT cloud and edge layers should be implemented to order to overcome this issue. 

### 5.5. Programming Networks for the Support of IoT Applications

The software-defined networking (SDN) [102] is a modern method that allows administrators to initialize, control, and manage the networking components of the OSI model [13]. The SDN was developed to overcome the issue and problems of the static architecture of traditional networks. The SDN has recently drawn performance in various fields such as bandwidth allocation, flow optimization, etc. Unfortunately, it has never been investigated with the use of the IoT. Thus, the researchers should focus on the study and the modification of traditional SDN protocols. 

### 5.6. Big Data Collection and Storage 

Data collection, processing, and analysis and visualization of data are very challenging tasks [36]. When we perform analysis and restrict it to a specific format, it may lead to a reduction in the efficiency of the results. Therefore, it is a very important task to have full knowledge of the IoT domain. It will help to decide the format and the structure of data collected by the sensors [103]. If we have less knowledge, then it means that it will result in garbage data. This will lead to an increase in the overall cost. The issue of five Vs occurs especially when dealing with big data in IoT technology [36].

### 5.7. Technical and Heterogeneity

As IoT end devices generate diverse types of data and have different communication protocols [20], these devices can have different shapes and sizes and are designed to communicate with others using cooperative applications [104]. Thus, to provide safety and authentication, the IoT system should assign a unique identification for each device.

## 6. Conclusions and Future Work

In this paper, we particularly analyzed and compared the most recent research studies related to the integration of big data, data science, and network science disciplines that enable researchers to understand from technical perspectives. We provided a conceptual view of the traditional IoT and advances that guide the researchers about this technology. We proposed a systematic catalog that covers the most recent advances in current IoT technology. We analyzed and compared different studies along with the weakness and strengths in different application areas. We provided additional coverage about leading technologies such as ML, AI, FL, and blockchain as feasible and promising solutions for the future IoT. From this, it was found that some research has already been conducted on these technologies and that these technologies are capable of addressing the current issue in the IoT. Based on the critical analysis of the recent literature, we have identified various challenges such as data provenance, integration and data management, etc. Deep learning and federated learning are still not investigated enough and need to be addressed in the near future by the research community.

## Figures and Tables

**Figure 1 sensors-22-08072-f001:**
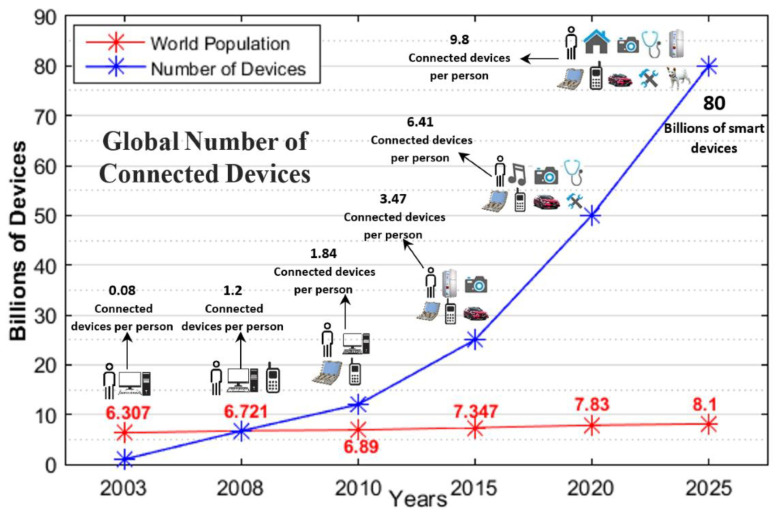
The current status and future perspective of the IoT.

**Figure 2 sensors-22-08072-f002:**
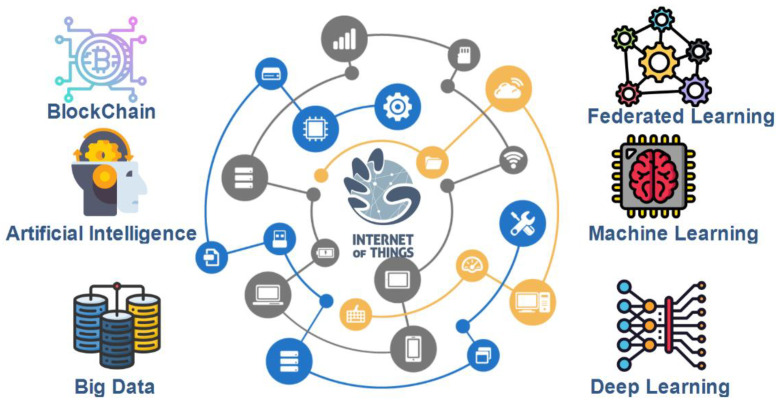
The connecting technologies in the IoT.

**Figure 3 sensors-22-08072-f003:**
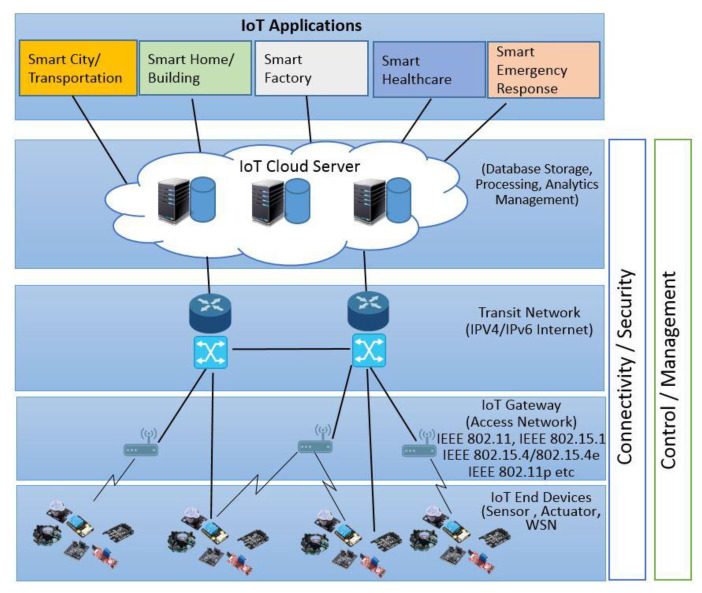
Conceptual view of the IoT.

**Figure 4 sensors-22-08072-f004:**
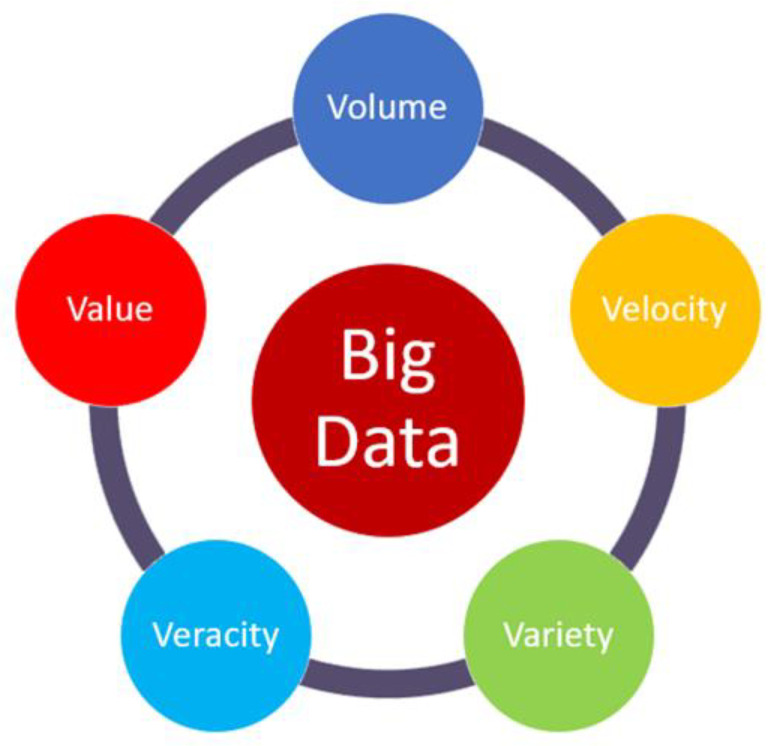
Five Vs of big data.

**Figure 5 sensors-22-08072-f005:**
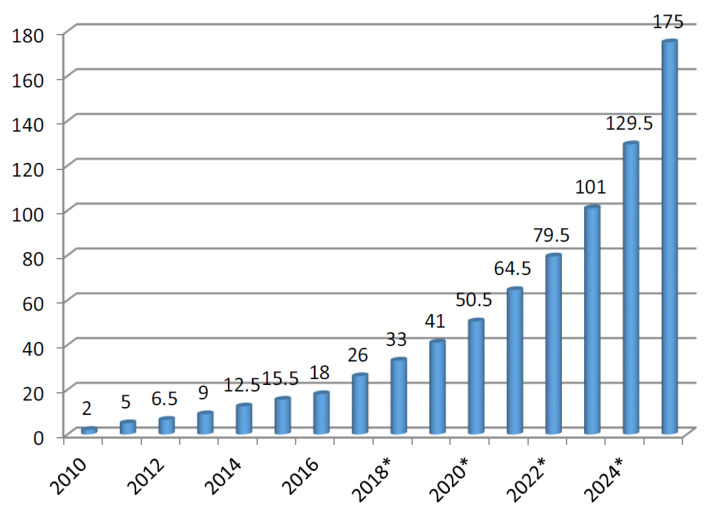
Data growth prediction dynamics worldwide.

**Figure 6 sensors-22-08072-f006:**
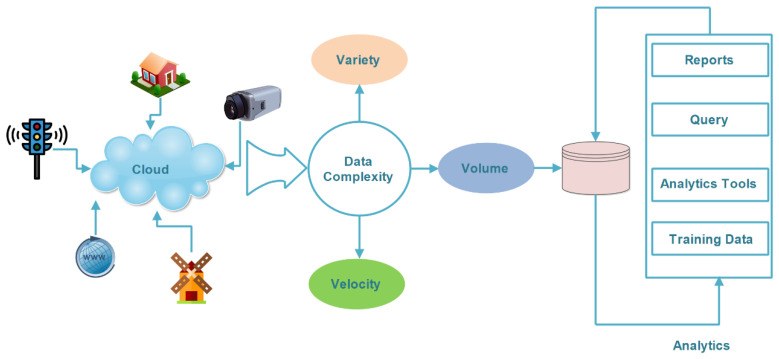
A holistic view of the IoT and big data analytics.

**Figure 7 sensors-22-08072-f007:**
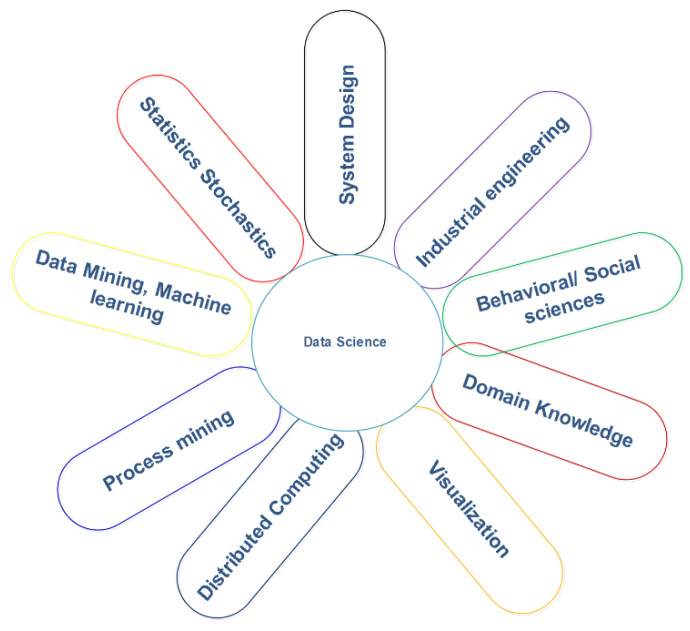
Data science.

**Figure 8 sensors-22-08072-f008:**
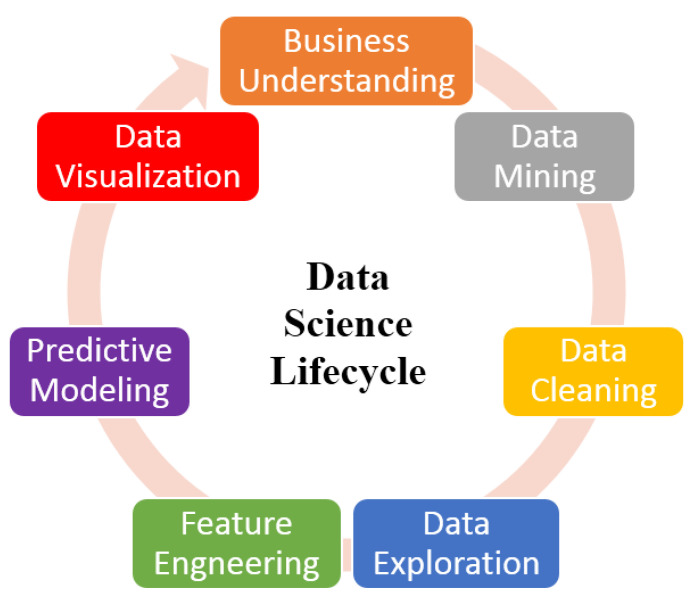
Data science lifecycle.

**Figure 9 sensors-22-08072-f009:**
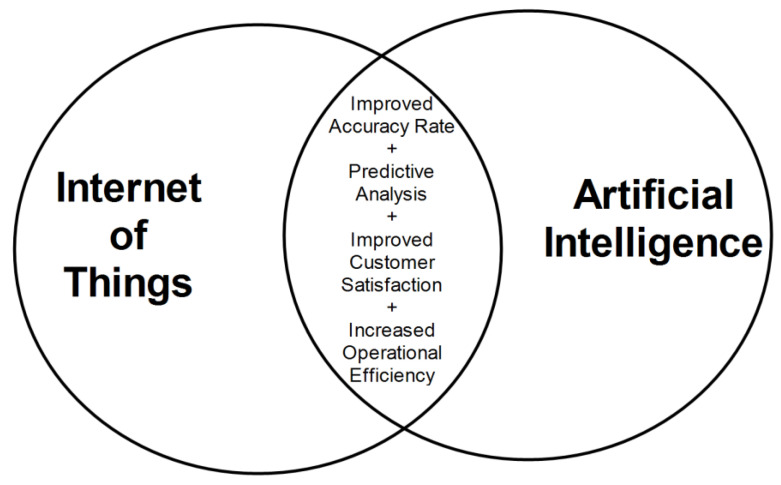
The common functionality in the IoT and artificial intelligence.

**Figure 10 sensors-22-08072-f010:**
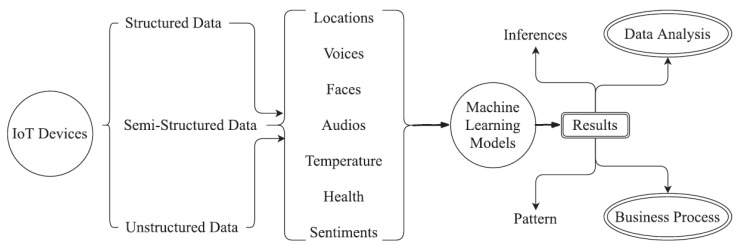
The basic machine learning-based model integration with the IoT.

**Table 1 sensors-22-08072-t001:** Summary and comparison of our survey and current recent research.

Reference Paper	Year	Big Data	Data Science	Network Science
Our Research	2022	☑	☑	☑
Amin et al. [5]	2021	⊠	☑	☑
Wang et al. [6]	2020	⊠	⊠	☑
Piccirilli et al. [7]	2020	⊠	☑	⊠
Youhua et al. [2]	2020	☑	**○**	⊠
Devi et al. [8]	2020	⊠	☑	⊠
Qiang et al. [10]	2019	○	☑	⊠
Ranjan et al. [11]	2018	⊠	☑	⊠
Foidl et al. [12]	2016	⊠	☑	⊠

☑ Covered. ⊠ Not Covered. ○ Briefly covered.

**Table 2 sensors-22-08072-t002:** The intelligent data processing methods in the IoT.

References	Years	Contribution
[47]	2022	An efficient energy-management and data-processing model using gateways has been proposed. This model is suitable for demands such as multi-tasking, inter-operability, classification, and fast data delivery between different modules.
[46]	2019	A deep learning model has been proposed for the fast data processing in IoT systems.
[45]	2017	The authors proposed a framework using multi-agents for cleaning and pre-processing of data in the IoT.
[48]	2017	The authors proposed a unique model for IoT-based smart healthcare. They had discussed the challenges and the state-of-the-art methods in healthcare systems.

## Data Availability

Not applicable.

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
