# Peer review of "A Step toward Next-Generation Advancements in the Internet of Things Technologies"

_sensors, 2022, doi:10.3390/s22208072_

Round 1

Reviewer 1 Report

I have the following comments on the content of the paper:

- figure 3 is from the year 2018 - are there any more up-to-date predictions since the predictions change from year to year?

- the 4th section discusses the main objective of the paper, but it does not consist of enough references and does not discuss the objective in a sufficiently detailed way and should be extended

- there is very little information about the actual themes like artificial intelligence, machine learning, blockchain, and federated learning -- and all of them are very tied up with the IoT and should be discussed in the 5th section thoroughly

- the conclusion consists of only the conclusion of the paper content but does not offer any discussion from the authors of the paper, nor does it include a discussion about the future work

I have the following comments on the formal aspect of the paper:

- I don't think there should be brackets in the title of the paper; please consider using the full Internet of Things or just the abbreviation (IoT) in the title

- formatting of table 1 should be corrected

- a lot of section names are on the end of the pages and should be corrected in the final paper

- referencing of the figures/tables should be corrected (Figure.XX.) in the whole paper (consider using \figurename~\ref{...} and \tablename~\ref{...})

- the figures are missing the references

- the figures should be of better quality, dpi should be higher, and some of them are too big (the text in the figures should be about the same as the text in the paper)

Reviewer 2 Report

The author presented a survey of next-generation IoT technologies from a data science perspective. This is interesting. However, I have several comments shown as follows.

1. First, there are a lot of related surveys for IoTs, and the authors should point out what is the main difference between them.

2. Big data and network science are two very broad concepts, please emphasize some more concrete aspects of IoT for data science.

3. Please give a logical figure or table to demonstrate the architecture of this paper.

4. Some expressions and figures in this paper are not professional, please improve them.

5. As a survey, the number of references is not enough, lack some recent references such as:

1) Mohammadi, M., Al-Fuqaha, A., Sorour, S., & Guizani, M. (2018). Deep learning for IoT big data and streaming analytics: A survey. IEEE Communications Surveys & Tutorials, 20(4), 2923-2960. 2) Sollins, K. R. (2019). IoT big data security and privacy versus innovation. IEEE Internet of Things Journal, 6(2), 1628-1635. 3) Guo, J., Ding, X., & Wu, W. (2021). Reliable traffic monitoring mechanisms based on blockchain in vehicular networks. IEEE Transactions on Reliability, 71 (3), 1219-1229. 4) Guo, J., Ding, X., & Wu, W. (2022). An Architecture for Distributed Energies Trading in Byzantine-Based Blockchains. IEEE Transactions on Green Communications and Networking, 6(2), 1216-1230.

Round 2

Reviewer 1 Report

I still think the paper should be more thorough in the 4th section where the authors describe the state-of-the-art technologies in IoT as per my previous comments (hence the major revision). I don't believe 63 referenced papers are enough to entitle the paper to the "review" category. 

The quality and cropping of some images should still be improved. 
